# AssembleNet: Searching for Multi-Stream Neural Connectivity in Video Architectures

**Michael S. Ryoo**[1,2]**, AJ Piergiovanni**[1,2,3]**, Mingxing Tan**[2] **& Anelia Angelova**[1,2]
[1]Robotics at Google
[2]Google Research
[3]Indiana University Bloomington
`mryoo@google.com`

## Abstract

Learning to represent videos is a very challenging task both algorithmically and computationally. Standard video CNN architectures have been designed by directly extending architectures devised for image understanding to include the time dimension, using modules such as 3D convolutions, or by using two-stream design to capture both appearance and motion in videos. We interpret a video CNN as a collection of multi-stream convolutional blocks connected to each other, and propose the approach of automatically finding neural architectures with better connectivity and spatio-temporal interactions for video understanding. This is done by evolving a population of overly-connected architectures guided by connection weight learning. Architectures combining representations that abstract different input types (i.e., RGB and optical flow) at multiple temporal resolutions are searched for, allowing different types or sources of information to interact with each other. Our method, referred to as AssembleNet, outperforms prior approaches on public video datasets, in some cases by a great margin. We obtain 58.6% mAP on Charades and 34.27% accuracy on Moments-in-Time.

## 1 Introduction

Learning to represent videos is a challenging problem. Because a video contains spatio-temporal data, its representation is required to abstract both appearance and motion information. This is particularly important for tasks such as activity recognition, as understanding detailed semantic contents of the video is needed. Previously, researchers approached this challenge by designing a two-stream model for appearance and motion information respectively, combining them by late or intermediate fusion to obtain successful results: Simonyan & Zisserman (2014); Feichtenhofer et al. (2016b;a; 2017; 2018). However, combining appearance and motion information is an open problem and the study on how and where different modalities should interchange representations and what temporal aspect/resolution each stream (or module) should focus on has been very limited.

In this paper, we investigate how to learn feature representations across spatial and motion visual clues. We propose a new multi-stream neural architecture search algorithm with *connection learning guided evolution*, which focuses on finding higher-level connectivity between network blocks taking multiple input streams at different temporal resolutions. Each block itself is composed of multiple residual modules with space-time convolutional layers, learning spatio-temporal representations. Our architecture learning not only considers the connectivity between such multi-stream, multi-resolution blocks, but also merges and splits network blocks to find better multi-stream video CNN architectures. Our objective is to address two main questions in video representation learning: (1) what feature representations are needed at each intermediate stage of the network and at which resolution and (2) how to combine or exchange such intermediate representations (i.e., connectivity learning). Unlike previous neural architecture search methods for images that focus on finding a good 'module' of convolutional layers to be repeated in a single-stream networks (Zoph et al., 2018; Real et al., 2019), our objective is to search for higher-level connections between multiple sequential or concurrent blocks to form multi-stream architectures.

We propose the concept of AssembleNet, a new method of fusing different sub-networks with different input modalities and temporal resolutions. AssembleNet is a general formulation that

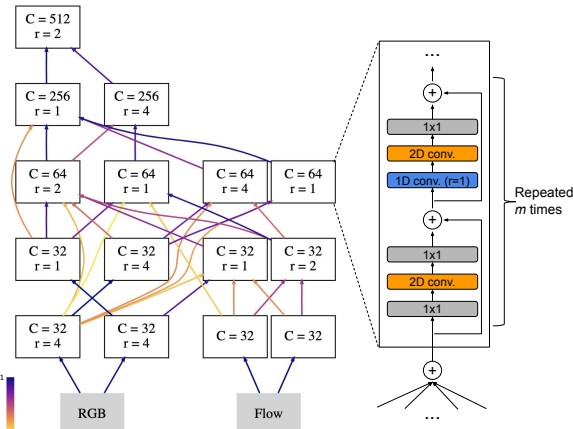

Figure 1: AssembleNet with multiple intermediate streams. Example learned architecture. Darker colors of connections indicate stronger connections. At each convolutional block, multiple 2D and (2+1)D residual modules are repeated alternatingly. Our network has 4 block levels (+ the stem level connected to raw data). Each convolutional block has its own output channel size (i.e., the number of filters) $C$ and the temporal resolution $r$ controlling the 1D temporal convolutional layers in it.

allows representing various forms of multi-stream CNNs as directed graphs, coupled with an efficient evolutionary algorithm to explore the network connectivity. Specifically, this is done by utilizing the learned connection weights to guide evolution, in addition to randomly combining, splitting, or connecting sub-network blocks. AssembleNet is a 'family' of learnable architectures; they provide a generic approach to learn connectivity among feature representations across input modalities, while being optimized for the target task. We believe this is the first work to (i) conduct research on automated architecture search with multi-stream connections for video understanding, and (ii) introduce the new connection-learning-guided evolutionary algorithm for neural architecture search.

Figure 1 shows an example learned AssembleNet. The proposed algorithm for learning video architectures is very effective: it outperforms all prior work and baselines on two very challenging benchmark datasets, and establishes a new state-of-the-art. AssembleNet models use equivalent number of parameters to standard two-stream (2+1)D ResNet models.

## 2 PREVIOUS WORK

A video is a spatio-temporal data (i.e., image frames concatenated along time axis), and its representation must abstract both spatial and temporal information. Full 3D space-time (i.e., XYT) convolutional layers as well as (2+1)D convolutional layers have been popularly used to represent videos (Tran et al., 2014; Carreira & Zisserman, 2017; Tran et al., 2018; Xie et al., 2018). Researchers studied replacing 2D convolutional layers in standard image-based CNNs such as Inception (Szegedy et al., 2016) and ResNet (He et al., 2016), so that it can be directly used for video classification.

Two-stream network designs, which combine motion and appearance inputs, are commonly used (e.g., Simonyan & Zisserman, 2014; Feichtenhofer et al., 2016a; 2017; 2016b). Combining appearance information at two different temporal resolutions (e.g., 24 vs. 3 frames per second) with intermediate connections has been proposed by Feichtenhofer et al. (2018). Late fusion of the two-stream representations or architectures with more intermediate connections (Diba et al., 2019), have also been explored. However, these video CNN architectures are the result of careful manual designs by human experts.

Neural Architecture Search (NAS), the concept of automatically finding better architectures based on data, is becoming increasingly popular (Zoph & Le, 2017; Zoph et al., 2018; Liu et al., 2018). Rather than relying on human expert knowledge to design a CNN model, neural architecture search allows the machines to generate better performing models optimized for the data. The use of reinforcement learning controllers (Zoph & Le, 2017; Zoph et al., 2018) as well as evolutionary algorithms (Real et al., 2019) have been studied, and they meaningfully outperform handcrafted architectures. Most of

these works focus on learning architectures of modules (i.e., groupings of layers and their connections) to be repeated within a fixed single-stream meta-architecture (e.g., ResNet) for image-based object classification. One-shot architecture search to learn differentiable connections (Bender et al., 2018; Liu et al., 2019) has also been successful for images. However, it is very challenging to directly extend such work to find multi-stream models for videos, as it requires preparing all possible layers and interactions the final architecture may consider using. In multi-stream video CNNs, there are many possible convolutional blocks with different resolutions, and fully connecting them requires a significant amount of memory and training data, which makes it infeasible.

Our work is also related to Ahmed & Torresani (2017) which used learnable gating to connect multiple residual module branches, and to the RandWire network (Xie et al., 2019), which showed that randomly connecting a sufficient number of convolutional layers creates performant architectures. However, similar to previous NAS work, the latter focuses only on generating connections between the layers within a block. The meta-architecture is fixed as a single stream model with a single input modality. In this work, our objective is to learn high-level connectivity between multi-stream blocks for video understanding driven by data. We confirm experimentally that in the multi-stream video CNNs, where multiple types of input modalities need to be considered at various resolutions, randomly connecting blocks is insufficient and the proposed architecture learning strategy is necessary.

## 3 ASSEMBLENET

We propose a new principled way to find better neural architectures for video representation learning. We first expand a video CNN to a multi-resolution, multi-stream model composed of multiple sequential and concurrent neural blocks, and introduce a novel algorithm to search for the optimal connectivity between the blocks for a given task.

We model a video CNN architecture as a collection of convolutional blocks (i.e., sub-networks) connected to each other. Each block is composed of a residual module of space-time convolutional layers repeated multiple times, while having its own temporal resolution. The objective of our video architecture search is to automatically (1) decide the number of parallel blocks (i.e., how many streams to have) at each level of the network, (2) choose their temporal resolutions, and (3) find the optimal connectivity between such multi-stream neural blocks across various levels. The highly interconnected convolutional blocks allow learning of the video representations combining multiple input modalities at various temporal resolutions. We introduce the concept of connection-learning-guided architecture evolution to enable multi-stream architecture search.

We name our final architecture as an 'AssembleNet', since it is formulated by assembling (i.e., merging, splitting, and connecting) multiple building blocks.

### 3.1 GRAPH FORMULATION

In order to make our neural architecture evolution consider multiple different streams with different modalities at different temporal resolutions, we formulate the multi-stream model as a directed acyclic graph. Each node in the graph corresponds to a sub-network composed of multiple convolutional layers (i.e., a block), and the edges specify the connections between such sub-networks. Each architecture is denoted as $G_i = (N_i, E_i)$ where $N_i = \{n_{0i}, n_{1i}, n_{2i}, \cdots\}$ is the set of nodes and $E_i$ is the set of edges defining their connectivity.

**Nodes.** A node in our graph representation is a ResNet block composed of a fixed number of interleaved 2D and (2+1)D residual modules. A '2D module' is composed of a 1x1 conv. layer, one 2D conv. layer with filter size 3x3, and one 1x1 convolutional layer. A '(2+1)D module' consists of a temporal 1D convolutional layer (with filter size 3), a 2D conv. layer, and a 1x1 conv. layer. In each block, we repeat a regular 2D residual module followed by the (2+1)D residual module $m$ times.

Each node has its own block level, which naturally decides the directions of the edges connected to it. Similar to the standard ResNet models, we made the nodes have a total of four block levels (+ the stem level). Having multiple nodes of the same level means the architecture has multiple parallel 'streams'. Figure 1 illustrates an example. Each level has a different $m$ value: 1.5, 2, 3, and 1.5. $m = 1.5$ means that there is one 2D module, one (2+1)D module, and one more 2D module. As a result, the depth of our network is 50 conv. layers. We also have a batch normalization layer followed by a ReLU after every conv. layer.

There are two special types of nodes with different layer configurations: source nodes and sink nodes. A source node in the graph directly takes the input and applies a small number of convolutional/pooling layers (it is often referred as the 'stem' of a CNN model). In video CNNs, the input is a 4D tensor (XYT + channel) obtained by concatenating either RGB frames or optical flow images along the time axis. Source nodes are treated as level-0 nodes. The source node is composed of one 2D conv. layer of filter size 7x7, one 1D temporal conv. layer of filter size 5, and one spatial max pooling layer. The 1D conv. is omitted in optical flow stems. A sink node generates the final output of the model, and it is composed of one pooling, one fully connected, and one softmax layer. The sink node is also responsible for combining the outputs of multiple nodes at the highest level, by concatenating them after the pooling. More details are provided in Appendix.

Each node in the graph also has two attributes controlling the convolutional block: its temporal resolution and the number of channels. We use temporally dilated 1D convolution to dynamically change the resolution of the temporal convolutional layers in different blocks, which are discussed more below. The channel size (i.e., the number of filters) of a node could take arbitrary values, but we constrain the sum of the channels of all nodes in the same block level to be a constant so that the capacity of an AssembleNet model is equivalent to a ResNet model with the same number of layers.

**Temporally Dilated 1D Convolution.**    One of the objectives is to allow the video architectures to look at multiple possible temporal resolutions. This could be done by preparing actual videos with different temporal resolutions as in Feichtenhofer et al. (2018) or by using temporally 'dilated convolutions as we introduce here. Having dilated filters allow temporal 1D conv. layers to focus on different temporal resolution without losing temporal granularity. This essentially is a 1D temporal version of standard 2D dilated convolutions used in Chen et al. (2018) or Yu & Koltun (2016):

Let $k$ be a temporal filter (i.e., a vector) with size $2d + 1$. The dilated convolution operator $*_r$ is similar to regular convolution but has different steps for the summation, described as:

$$(F *_r k)(t) = \sum_{t_1 + rt_2 = t} F(t_1)k(t_2 + d) \tag{1}$$

where $t$, $t_1$, and $t_2$ are time indexes. $r$ indicates the temporal resolution (or the amount of dilation), and the standard 1D temporal convolution is a special case where $r = 1$. In the actual implementation, this is done by inserting $r - 1$ number of zeros between each element of $k$ to generate $k'$, and then convolving such zero-inflated filters with the input: $F *_r k = F * k'$. Importantly, the use of the dilated convolution allows different intermediate sub-network blocks (i.e., not just input stems) to focus on very different temporal resolutions at different levels of the convolutional architecture.

Note that our temporally dilated convolution is different from the one used in Lea et al. (2017), which designed a specific layer to combine representations from different frames with various step sizes. Our layers dilate the temporal filters themselves. Our dilated convolution can be viewed as a direct temporal version of the standard dilated convolutions used in Chen et al. (2018); Yu & Koltun (2016).

**Edges.**    Each directed edge specifies the connection between two sub-network blocks, and it describes how a representation is transferred from one block to another block. We constrain the direction of each edge so that it is connected from a lower level block to a higher level block to avoid forming a cycle and allow parallel streams. A node may receive inputs from any number of lower-level nodes (including skip connections) and provide its output to any number of higher-level nodes.

Our architectures use a (learnable) weighted summation to aggregate inputs given from multiple connected nodes. That is, an input to a node is computed as $F^{in} = \sum_i \text{sigmoid}(w_i) \cdot F_i^{out}$, where $F_i^{out}$ are output tensors (i.e., representations) of the nodes connected to the node and $w_i$ are their corresponding weights. Importantly, each $w_i$ is considered as a variable that has to be learned from training data through back propagation. This has two key advantages compared to conventional feature map concatenation: (i) The input tensor size is consistent regardless of the number of connections. (ii) We use learned connection weights to 'guide' our architecture evolution algorithm in a preferable way, which we discuss more in Section 3.2.

If the inputs from different nodes differ in their spatial size, we add spatial max pooling and striding to match their spatial size. If the inputs have different channel sizes, we add a 1x1 conv. layer to match the bigger channel size. Temporal sizes of the representations is always consistent in our graphs, as there is no temporal striding in our formulation and the layers in the nodes are fully convolutional.

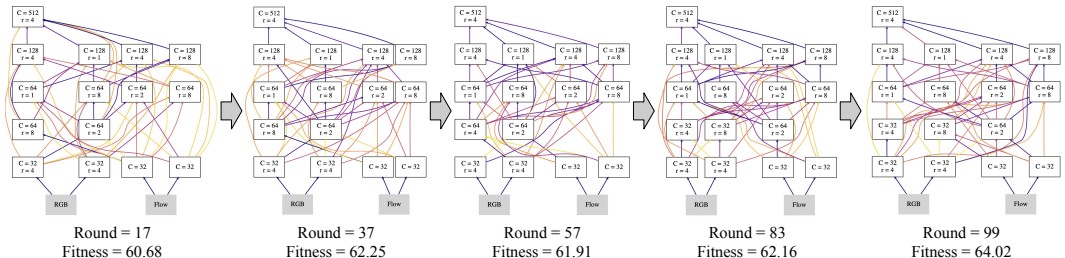

Round = 17
Fitness = 60.68

Round = 37
Fitness = 62.25

Round = 57
Fitness = 61.91

Round = 83
Fitness = 62.16

Round = 99
Fitness = 64.02

Figure 2: An example showing a sequence of architecture evolution. These architectures have actual parent-child relationships. The fitness of the third model was worse than the second model (due to random mutations), but it was high enough to survive in the population pool and eventually evolve into a better model.

## 3.2 EVOLUTION

We design an evolutionary algorithm with discrete mutation operators that modify nodes and edges in architectures over iterations. The algorithm maintains a population of $P$ different architectures, $P = \{G_1, G_2, \cdots, G_{|P|}\}$, where each architecture $G$ is represented with a set of nodes and their edges as described above.

The initial population is formed by preparing a fixed number of randomly connected architectures (e.g., $|P| = 20$). Specifically, we (1) prepare a fixed number of stems and nodes at each level (e.g., two per level), (2) apply a number of node split/merge mutation operators which we discuss more below, and (3) randomly connect nodes with the probability $p = 0.5$ while discarding architectures with graph depth $< 4$. As mentioned above, edges are constrained so that there is no directed edge reversing the level ordering. Essentially, a set of overly-connected architectures are used as a starting point. Temporal resolutions are randomly assigned to the nodes.

We use the tournament selection algorithm (Goldberg & Deb, 1991) as the main evolution framework: At each evolution round, the algorithm updates the population by selecting a 'parent' architecture and mutating (i.e., modifying) it to generate a new 'child' architecture. The parent is selected by randomly sampling a subset of the entire population $P' \subset P$, and then computing the architecture with the highest 'fitness': $G_p = \mathrm{argmax}_{G_i \in P'} f(G_i)$ where $f(G)$ is the fitness function. Our fitness is defined as a video classification accuracy of the model, measured by training the model with a certain number of initial iterations and then evaluating it on the validation set as its proxy task. More specifically, we use top-1 accuracy + top-5 accuracy as the fitness function. The child is added into the population, and the model with the least fitness is discarded from the population.

A child is evolved from the parent by following two steps. First, it changes the block connectivity (i.e., edges) based on their learned weights: 'connection-learning-guided evolution'. Next, it applies a random number of mutation operators to further modify the node configuration. The mutation operators include (1) a random modification of the temporal resolution of a convolutional block (i.e., a node) as well as (2) a merge or split of a block. When splitting a node into two nodes, we make their input/output connections identical while making the number of channels in their convolutional layers half that of the node before the split (i.e., $C = C_p/2$ where $C_p$ is the channel size of the parent). More details are found in Appendix. As a result, we maintain the total number of parameters, since splitting or merging does not change the number of parameters of the convolutional blocks.

**Connection-Learning-Guided Mutation.** Instead of randomly adding, removing or modifying block connections to generate the child architecture, we take advantage of the learned connection weights from its parent architecture. Let $E_p$ be the set of edges of the parent architecture. Then the edges of the child architecture $E_c$ are inherited from $E_p$, by only maintaining high-weight connections while replacing the low-weight connections with new random ones. Specifically, $E_c = E_c^1 \cup E_c^2$:

$$E_c^1 = \left\{ e \in E_p \mid W_e > B \right\}, \quad E_c^2 = \left\{ e \in (E_* - E_p) \mid \frac{|E_p - E_c^1|}{|E - E_p|} > X_e \right\} \quad (2)$$

where $X \sim \mathrm{unif}(0,1)$ and $E_*$ is the set of all possible edges. $E_c^1$ corresponds to the edges the child architecture inherits from the parent architecture, decided based on the learned weight of the edge $W_e$.

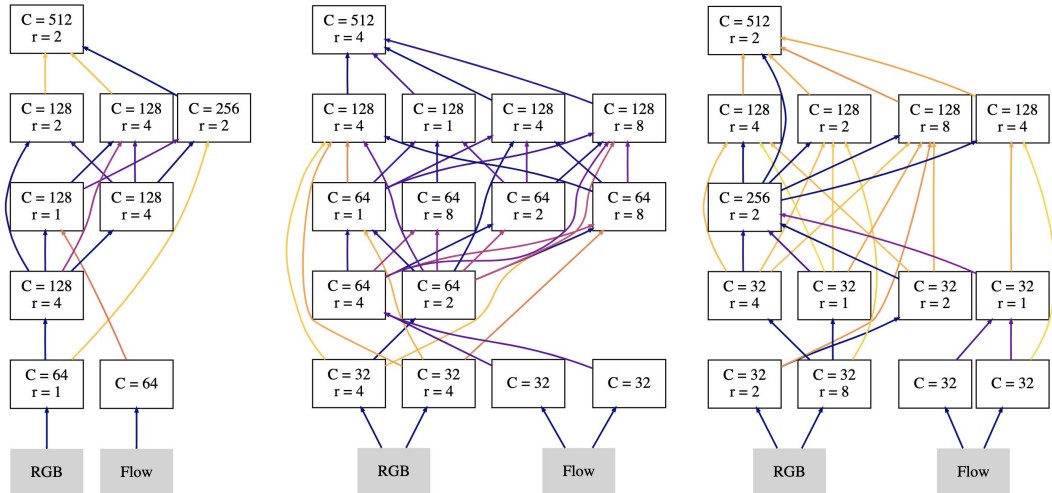

Figure 3: More AssembleNet examples. Similarly good performing diverse architectures, all with higher-than-50% mean-average precision on Charades. For instance, even our simpler two-stem AssembleNet-50 (left) got 51.4% mAP on Charades. Darker edges mean higher weights.

This is possible because our fitness measure involves initial proxy training of each model, providing the learned connection weight values $W_e$ of the parent.

$B$, which controls whether or not to keep an edge from the parent architecture, could either be a constant threshold or a random variable following a uniform distribution: $B = b$ or $B = X_B \sim$ unif$(0, 1)$. $E_c^2$ corresponds to the new randomly added edges which were not in the parent architecture. We enumerate through each possible new edge, and randomly add it with the probably of $|E_p - E_c^1|/|E - E_p|$. This makes the expected total number of added edges to be $|E_p - E_c^1|$, maintaining the size of $E_c$. Figure 2 shows an example of the evolution process and Figure 3 shows final architectures.

**Evolution Implementation Details.** Initial architectures are formed by randomly preparing either {2 or 4} stems, two nodes per level at levels 1 to 3, and one node at level 4. We then apply 1∼5 random number of node split operators so that each initial architecture has a different number of nodes. Each node is initialized with a random temporal resolution of 1, 2, 4, or 8. As mentioned, each possible connection is then added with the probability of $p = 0.5$.

At each evolution round, the best-performing parent architecture is selected from a random subset of 5 from the population. The child architecture is generated by modifying the connections from the parent architecture (Section 3.2). A random number of node split, merge, or temporal resolution change mutation operators (0∼4) are then applied. Evaluation of each architecture (i.e., measuring the fitness) is done by training the model for 10K iterations and then measuring its top-1 + top-5 accuracy on the validation subset. The Moments-in-Time dataset, described in the next section, is used as the proxy dataset to measure fitness. The evolution was run for ∼200 rounds, although a good performing architecture was found within only 40 rounds (e.g., Figure 3-right). Figure 1 shows the model found at the 165th round. 10K training iterations of each model during evolution took 3∼5 hours; with our setting, evolving a model for 40 rounds took less than a day with 10 parallel workers.

## 4 EXPERIMENTS

### 4.1 DATASETS

**Charades Dataset.** We first test on the popular Charades dataset (Sigurdsson et al., 2016) which is unique in the activity recognition domain as it contains long sequences. It is one of the largest public datasets with continuous action videos, containing 9848 videos of 157 classes (7985 training and 1863 testing videos). Each video is ∼30 seconds. It is a challenging dataset due to the duration and variety of the activities. Activities may temporally overlap in a Charades video, requiring the model to predict multiple class labels per video. We used the standard 'Charades_v1_classify' setting for the evaluation. To comply with prior work (e.g. Feichtenhofer et al., 2018), we also report results when pre-training on Kinetics (Carreira & Zisserman, 2017), which is another large-scale dataset.

Table 1: Reported state-of-the-art action classification performances (vs. AssembleNet) on Charades. '2-stream (2+1)D ResNet-50' is the two-stream model with connection learning for level-4 fusion.

| Method | pre-train | modality | mAP |
|---|---|---|---|
| 2-stream (Simonyan & Zisserman, 2014) | UCF101 | RGB+Flow | 18.6 |
| Asyn-TF (Sigurdsson et al., 2017) | UCF101 | RGB+Flow | 22.4 |
| CoViAR (Wu et al., 2018b) | ImageNet | Compressed | 21.9 |
| MultiScale TRN (Zhou et al., 2018) | ImageNet | RGB | 25.2 |
| I3D (Carreira & Zisserman, 2017) | Kinetics | RGB | 32.9 |
| I3D (from Wang et al., 2018) | Kinetics | RGB | 35.5 |
| I3D-NL (Wang et al., 2018) | Kinetics | RGB | 37.5 |
| STRG (Wang & Gupta, 2018) | Kinetics | RGB | 39.7 |
| LFB-101 (Wu et al., 2018a) | Kinetics | RGB | 42.5 |
| SlowFast-101 (Feichtenhofer et al., 2018) | Kinetics | RGB+RGB | 45.2 |
| 2-stream (2+1)D ResNet-50 (ours) | MiT | RGB+Flow | 48.7 |
| 2-stream (2+1)D ResNet-50 (ours) | Kinetics | RGB+Flow | 50.4 |
| 2-stream (2+1)D ResNet-101 (ours) | Kinetics | RGB+Flow | 50.6 |
| AssembleNet-50 (ours) | MiT | RGB+Flow | 53.0 |
| AssembleNet-50 (ours) | Kinetics | RGB+Flow | 56.6 |
| AssembleNet-101 (ours) | Kinetics | RGB+Flow | **58.6** |

Table 2: State-of-the-art action classification accuracies on Moments in Time (Monfort et al., 2018).

| Method | modality | Top-1 | Top-5 |
|---|---|---|---|
| ResNet50-ImageNet | RGB | 27.16 | 51.68 |
| TSN (Wang et al., 2016) | RGB | 24.11 | 49.10 |
| Ioffe & Szegedy (2015) | Flow | 11.60 | 27.40 |
| TSN-Flow (Wang et al., 2016) | Flow | 15.71 | 34.65 |
| TSN-2Stream (Wang et al., 2016) | RGB+F | 25.32 | 50.10 |
| TRN-Multi (Zhou et al., 2018) | RGB+F | 28.27 | 53.87 |
| Two-stream (2+1)D ResNet-50 | RGB+F | 28.97 | 55.55 |
| I3D (Carreira & Zisserman, 2017) | RGB+F | 29.51 | 56.06 |
| AssembleNet-50 | RGB+F | 31.41 | 58.33 |
| AssembleNet-50 (with Kinetics) | RGB+F | 33.91 | 60.86 |
| AssembleNet-101 (with Kinetics) | RGB+F | **34.27** | **62.71** |

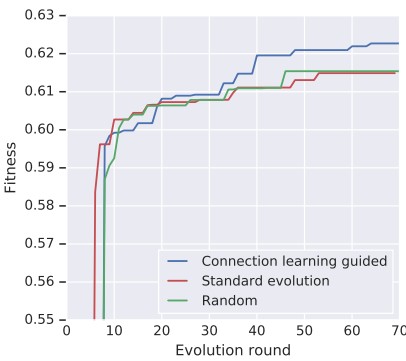

Figure 4: Comparison of different search methods.

We note that Kinetics is shrinking in size (∼15% videos removed from the original Kinetics-400) and the previous versions are no longer available from the official site.

**Moments in Time (MiT) Dataset.** The Moments in Time (MiT) dataset (Monfort et al., 2018) is a large-scale video classification dataset with more than 800K videos (∼3 seconds per video). It is a very challenging dataset with the state-of-the-art models obtaining less than 30% accuracy. We use this dataset for the architecture evolution, and train/test the evolved models. We chose the MiT dataset because it provides a sufficient amount of training data for video CNN models and allows stable comparison against previous models. We used its standard classification evaluation setting.

## 4.2 RESULTS

Tables 1 and 2 compare the performance of AssembleNet against the state-of-the-art models. We denote AssembleNet more specifically as AssembleNet-50, since its depth is 50 layers and has an equivalent number of parameters to ResNet-50. AssembleNet-101 is its 101 layer version having equivalent number of parameters to ResNet-101. AssembleNet is outperforming prior works on both datasets, setting new state-of-the-art results for them. Its performance on MiT is the first above 34%. We also note that the performances on Charades is even more impressive at 58.6 whereas previous known best results are 42.5 and 45.2. For these experiments, the architecture search was done on the MiT dataset, and then the found models are trained and tested on both datasets, which demonstrates that the found architectures are useful across datasets.

Table 3: Comparison between AssembleNet and architectures without evolution, but with connection weight learning. Four-stream models are reported here for the first time, and are very effective. All these models have a similar number of parameters.

| Architecture | MiT | Charades |
|---|---|---|
| Two-stream (late fusion) | 28.97 | 46.5 |
| Two-stream (fusion at lv. 4) | 30.00 | 48.7 |
| Two-stream (flow→RGB inter.) | 30.21 | 49.5 |
| Two-stream (fully, fuse at 4) | 29.87 | 50.5 |
| Four-stream (fully, fuse at 4) | 29.98 | 50.7 |
| Random (+ connection learning) | 29.91 | 50.1 |
| AssembleNet-50 | 31.41 | 53.0 |

Table 4: Ablation comparing different AssembleNet architectures found with full vs. constrained search spaces. The models are trained from scratch.

| Architecture | MiT |
|---|---|
| Baseline (random + conn. learning) | 29.91 |
| No mutation | 30.26 |
| RGB-only | 30.30 |
| Without temporal dilation | 30.49 |
| Two-stem only | 30.75 |
| Full AssembleNet-50 | 31.41 |

In addition, we compare the proposed connection-learning-guided evolution with random architecture search and the standard evolutionary algorithm with random connection mutations. We made the standard evolutionary algorithm randomly modify 1/3 of the total connections at each round, as that is roughly the number of edges the connection-learning-guided evolution modifies. Figure 4 shows the results, visualizing the average fitness score of the three top-performing models in each pool. We observe that the connection-learning-guided evolution is able to find better architectures, and it is able to do so more quickly. The standard evolution performs similarly to random search and is not as effective. We believe this is due to the large search space the approach is required to handle, which is exponential to the number of possible connections. For instance, if there are $N$ nodes, the search space complexity is $2^{O(N^2)}$ just for the connectivity search. Note that the initial ~30 rounds are always used for random initialization of the model population, regardless of the search method.

## 4.3 ABLATION STUDIES

We conduct an ablation study comparing the evolved AssembleNet to multiple (2+1)D two-stream (or multi-stream) architectures which are designed to match the abilities of Assemblenet but without evolution. We note that these include very strong architectures that have not been explored before, such as the four-stream model with dense intermediate connectivity. We design competitive networks having various connections between streams, where the connection weights are also learned (see the supplementary material for detailed descriptions and visualizations). Note that all these models have equivalent capacity (i.e., number of parameters). The performance difference is due to network structure. Table 3 shows the results, demonstrating that these architectures with learnable interconnectivity are very powerful themselves and evolution is further beneficial. The Moments in Time models were trained from scratch, and the Charades models were pre-trained on MiT. In particular, we evaluated an architecture with intermediate connectivity from the flow stream to RGB, inspired by Feichtenhofer et al. (2016b; 2018) (+ connection weight learning). It gave 30.2% accuracy on MiT and 49.5% on Charades, which are not as accurate as AssembleNet. Randomly generated models (from 50 rounds of search) are also evaluated, confirming that such architectures do not perform well.

Further, we conduct another ablation to confirm the effectiveness of our search space. Table 4 compares the models found with our full search space vs. more constrained search spaces, such as only using two stems and not using temporal dilation (i.e., fixed temporal resolution).

## 4.4 GENERAL FINDINGS

As the result of connection-learning-guided architecture evolution, non-obvious and non-intuitive connections are found (Figure 3). As expected, more than one possible "connectivity" solution can yield similarly good results. Simultaneously, models with random connectivity perform poorly compared to the found AssembleNet. Our observations also include: (1) The models prefer to have only one block at the highest level, although we allow the search to consider having more than one block at that level. (2) The final block prefers simple connections gathering all outputs of the blocks in the 2nd to last level. (3) Many models use multiple blocks with different temporal resolutions at the same level, justifying the necessity of the multi-stream architectures. (4) Often, there are 1 or 2 blocks heavily connected to many other blocks. (5) Architectures prefer using more than 2 streams, usually using 4 at many levels.

## 5 CONCLUSION

We present AssembleNet, a new approach for neural architecture search using connection-learning-guided architecture evolution. AssembleNet finds multi-stream architectures with better connectivity and temporal resolutions for video representation learning. Our experiments confirm that the learned models significantly outperform previous models on two challenging benchmarks.

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

# A APPENDIX

## A.1 SUPER-GRAPH VISUALIZATION OF THE CONNECTIVITY SEARCH SPACE

Figure 5 visualizes all possible connections and channel/temporal resolution options our architecture evolution is able to consider. The objective of our evolutionary algorithm could be interpreted as finding the optimal sub-graph (of this super-graph) that maximizes the performance while maintaining the number of total parameters. Trying to directly fit such entire super-graph into the memory was infeasible in our experiments.

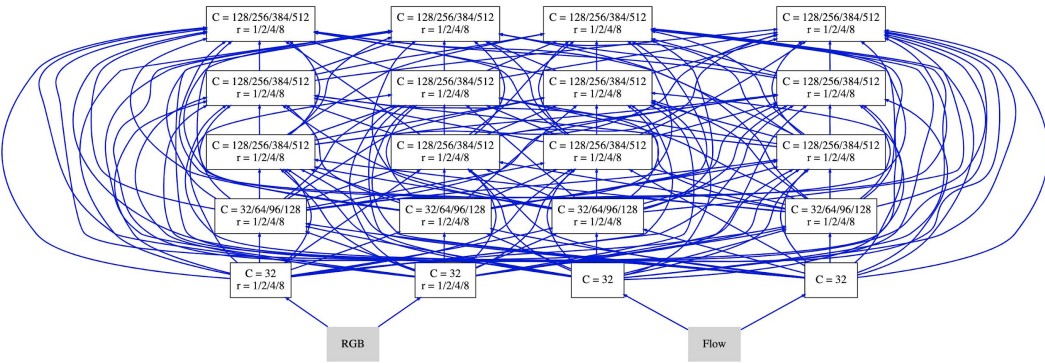

Figure 5: Visualization of the super-graph corresponding to our video architecture search space.

## A.2 CHANNEL SIZES OF THE LAYERS AND NODE SPLIT/MERGE MUTATIONS

As we described in the paper, each node (i.e., a convolutional block) has a parameter $C$ controlling the number of filters of the convolutional layers in the block. When splitting or merging blocks, the number of filters are split or combined respectively. Figure 6 provides a visualization of a block with number of filter specified to the right and a split operation. While many designs are possible, we design the blocks and splitting as follows. The size of 1x1 convolutional layers and 1D temporal convolutional layers are strictly governed by $C$, having the channel size of $C$ (some $4C$). On the other hand, the number of 2D convolutional layer is fixed per level as a constant $D_v$ where $v$ is the level of the block. $D_1 = 64$, $D_2 = 128$, $D_3 = 256$, and $D_4 = 512$. The layers in the stems have 64 channels if there are only two stems and 32 if there are four stems.

When a node is split into two nodes, we update the resulting two nodes' channel sizes to be half of their original node. This enables us to maintain the total number of model parameters before and after the node split to be identical. The first 1x1 convolutional layer will have half the parameters after the split, since its output channel size is now 1/2. The 2D convolutional layer will also have exactly half the parameters, since its input channel size is 1/2 while the output channel size is staying fixed. The next 1x1 convolutional layer will have the fixed input channel size while the output channel size becomes 1/2: thus the total number of parameters would be 1/2 of the original parameters.

Merging of the nodes is done in an inverse of the way we split. When merging two nodes into one, the merged node inherits all input/output connections from the two nodes: we take a union of all the connections. The channel size of the merged node is the sum of the channel sizes of the two nodes being merged. The temporal dilation rate of the merged node is randomly chosen between the two nodes before the merge.

## A.3 HAND-DESIGNED MODELS USED IN THE ABLATION STUDY

Figure 7 illustrates the actual architectures of the hand-designed (2+1)D CNN models used in our ablation study. We also show the final learned weights of the connections, illustrating which connections the model ended up using or not using. We note that these architectures are also very enlightening as the connectivity within them are learned in the process. We observe that stronger connections tend to be formed later for 2-stream architectures. For 4-stream architectures, stronger connections do form early, and, not surprisingly, a connection to at least one node of a different

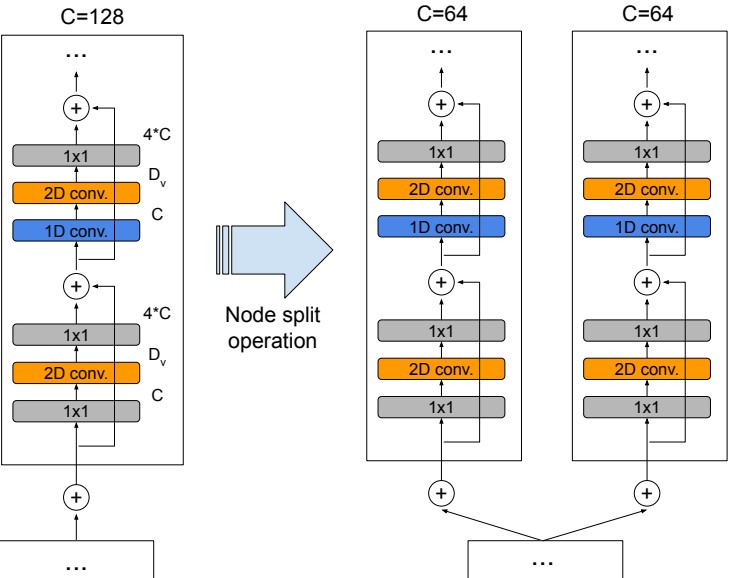

Figure 6: An illustration of the node split mutation operator, used for both evolution and initial architecture population generation.

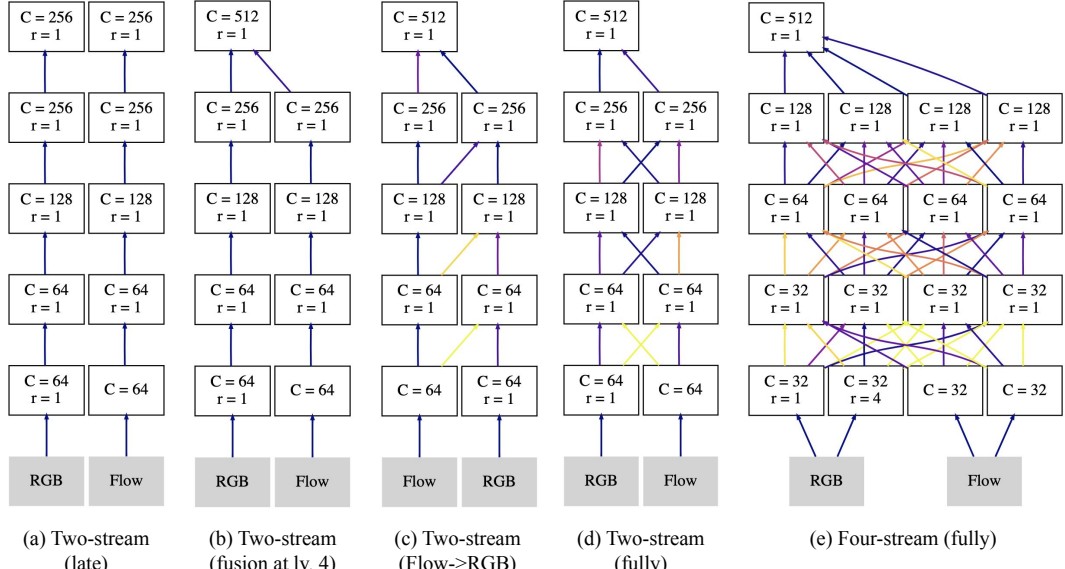

Figure 7: Illustration of hand-designed baseline (2+1)D CNN models used in our ablation study.

modality is established, i.e. a node stemming from RGB will connect to at least one flow node at the next level and vice versa.

Below is a more detailed description of the networks used in the paper: "Two-stream (late fusion)" means that the model has two separate streams at every level including the level 4, and the outputs of such two level 4 nodes are combined for the final classification. "Fusion at lv. 4" is the model that only has one level 4 node to combine the outputs of the two level 3 nodes using a weighted summation. "Two-stream (fully)" means that the model has two nodes at each level 1-3 and one node at level 4, and each node is always connected to every node in the immediate next level. "Flow→RGB" means that only the RGB stream nodes combine outputs from both RGB and flow stream nodes of the immediate lower level.

Table 5: The table form of the AssembleNet model with detailed parameters. This model corresponds to Figure 1. The parameters correspond to {node_level, input_node_list, $C$, $r$, and spatial stride}

| Index | Block parameters |
|-------|------------------|
| 0 | 0, [RGB], 32, 4, 4 |
| 1 | 0, [RGB], 32, 4, 4 |
| 2 | 0, [Flow], 32, 1, 4 |
| 3 | 0, [Flow], 32, 1, 4 |
| 4 | 1, [1], 32, 1, 1 |
| 5 | 1, [0], 32, 4, 1 |
| 6 | 1, [0,1,2,3], 32, 1, 1 |
| 7 | 1, [2,3], 32, 2, 1 |
| 8 | 2, [0, 4, 5, 6, 7], 64, 2, 2 |
| 9 | 2, [0, 2, 4, 7], 64, 1, 2 |
| 10 | 2, [0, 5, 7], 64, 4, 2 |
| 11 | 2, [0, 5], 64, 1, 2 |
| 12 | 3, [4, 8, 10, 11], 256, 1, 2 |
| 13 | 3, [8, 9], 256, 4, 2 |
| 14 | 4, [12, 13], 512, 2, 2 |

## A.4 ASSEMBLENET MODEL/LAYER DETAILS

We also provide the final AssembleNet model in table form in Table 5. In particular, the 2nd element of each block description shows the list of where the input to that block is coming from (i.e., the connections). As already mentioned, 2D and (2+1)D residual modules are repeated in each block. The number of repetitions $m$ are 1.5, 2, 3, and 1.5 at each level. $m = 1.5$ means that we have one 2D residual module, one (2+1)D module, and one more 2D module. This makes the number of convolutional layers of each block at levels 1-4 to be 9, 12, 18, and 9. In addition, a stem has at most 2 convolutional layers. The total depth of our network is 50, similar to a conventional (2+1)D ResNet-50. For AssembleNet-101, we use $m = 1.5$, 2, 11.5, and 1.5 at each level.

If a block has a spatial stride of 2, the striding happens at the first 2D convolutional layer of the block. In the stem which has the spatial stride of 4, the striding of size 2 happens twice, once at the 2D convolutional layer and at the max pooling layer. As mentioned, the model has a batch normalization layer followed by ReLU after every convolutional layer regardless of its type (i.e., 2D, 1D, and 1x1). 2D conv. filter sizes are 3x3, and 1D conv. filter sizes are 3.

## A.5 SINK NODE DETAILS

When each evolved or baseline (2+1)D model is applied to a video, it generates a 5D (BTYXC) tensor after the final convolutional layer, where B is the size of the batch and C is the number of channels. The sink node is responsible for mapping this into the output vector, whose dimensionality is identical to the number of video classes in the dataset. The sink node first applies a spatial average pooling to generate a 3D (BTC) tensor. If there are multiple level 4 nodes (which rarely is the case), the sink node combines them into a single tensor by averaging/concatenating them. Averaging or concatenating does not make much difference empirically. Next, temporal average/max pooling is applied to make the representation a 2D (BC) tensor (average pooling was used for the MiT dataset and max pooling was used for Charades), and the final fully connected layer and the soft max layer is applied to generate the final output.

## A.6 TRAINING DETAILS

For the Moments in Time (MiT) dataset training, 8 videos are provided per TPU core (with 16GB memory): the total batch size (for each gradient update) is 512 with 32 frames per video. The batch size used for Charades is 128 with 128 frames per video. The base framerate we used is 12.5 fps for MiT and 6 fps for Charades. The spatial input resolution is 224x224 during training. We used the standard Momentum Optimizer in TensorFlow. We used a learning rate of 3.2 (for MiT) and 25.6 (for Charades), 12k warmup iterations, and cosine decay. No dropout is used, weight decay is set to 1e-4 and label smoothing set to 0.2.

Training a model for 10k iterations (during evolution) took 3∼5 hours and fully training the model (for 50k iterations) took ∼24 hours per dataset.

We used the TV-L1 optical flow extraction algorithm (Zach et al., 2007) implemented with tensor operations by Piergiovanni & Ryoo (2019) to obtain flow input.

## A.7 EVALUATION DETAILS

When evaluating a model on the MiT dataset, we provide 36 frames per video. The duration of each MiT video is 3 seconds, making 36 frames roughly correspond to the entire video. For the Charades dataset where each video duration is roughly ∼30 seconds, the final class labels are obtained by applying the model to five random 128 frame crops (i.e., segments) of each video. The output multi-class labels are max-pooled to get the final label, and is compared to the ground truth to measure the average precision scores. The spatial resolution used for the testing is 256x256.

