# OpenReview forum: "AssembleNet: Searching for Multi-Stream Neural Connectivity in Video Architectures"
_ICLR.cc/2020/Conference — Accept (Poster)_

### Official Review · AnonReviewer2 · 2019-10-17
**Official Blind Review #2**

**Rating:** 8

**Review:**

This submission proposes a way to do multi-stream neural architecture search for video classification. I give an initial rating of accept because (1) there are not many work on video architecture search yet (2) the paper is well written (3) experiments are complete and results are strong. I have a few comments as below.

1. Most work in video action recognition tend not to use optical flow. Many people believe that if 3D conv or (2+1)D conv can be trained well, there is no point in using optical flow. What is the motivation of using flow in this work? I'm interested to know.

2. As shown in Figure 5 of the appendix, the search space is quite large. For each block, it seems that authors search with r=1/2/4/8. However, the best searched network seems to only has r=1, 2 or 4. This is kind of counter-intuitive because longer sequences should in general give better results. Is there an explanation or insight that why r=8 does not show up?

3. The learning rate for both datasets are very high, one is 3.2, the other is 25.6. This is quite unusual. Although many NAS literature show that large learning rate can help to achieve better performance, but 25.6 is really high. Did the authors do learning rate search as well? And what is the search space for learning rate?

4. I understand that this paper focus on learning the connectivity pattern from difference inputs, like rgb and flow. Have the authors tried using RGB alone and searching the architecture?



**Experience Assessment:**

I have published in this field for several years.

**Review Assessment: Checking Correctness Of Derivations And Theory:**

I carefully checked the derivations and theory.

**Review Assessment: Checking Correctness Of Experiments:**

I carefully checked the experiments.

**Review Assessment: Thoroughness In Paper Reading:**

I read the paper thoroughly.

---

> ### Author Response · Authors · 2019-11-12
> **We thank the reviewer for the comments and questions.**
>
> We thank the reviewer for the comments and questions. Please find our answers to the comments below:
>
> 1. "Most work in video action recognition tend not to use optical flow. Many people believe that if 3D conv or (2+1)D conv can be trained well, there is no point in using optical flow. What is the motivation of using flow in this work? I'm interested to know."
>
> Our motivation is that the explicit iterative optimization process in the optical flow extraction could allow the model to more easily capture motion information. This is also motivated by the observations of previous papers (e.g., I3D) suggesting the use of two-stream RGB+OpticalFlow models helps video recognition in practice. Training a pure 3D or (2+1)D conv layers to replicate the same behavior would likely require more layers and therefore more training data, and we wanted to overcome it given a fixed amount of data.
>
> 2. "As shown in Figure 5 of the appendix, the search space is quite large. For each block, it seems that authors search with r=1/2/4/8. However, the best searched network seems to only has r=1, 2 or 4. This is kind of counter-intuitive because longer sequences should in general give better results. Is there an explanation or insight that why r=8 does not show up?"
>
> This is because, while r=8 is allowed, it corresponds to a very large temporal stride which is more useful for videos on the longer side, which are not often seen in the dataset we used for the evolution (i.e., Moments-in-Time which has 3 second videos).
>
> 3. "The learning rate for both datasets are very high, one is 3.2, the other is 25.6. This is quite unusual. Although many NAS literature show that large learning rate can help to achieve better performance, but 25.6 is really high. Did the authors do learning rate search as well? And what is the search space for learning rate?"
>
> We checked multiple learning rates, starting with a base of 0.8 and increasing by a multiple of the batch size. While these learning rates are larger than for other tasks, this is likely due to a large batch size. We found a larger learning rate to be quite helpful for learning, and did not cause instability or other adverse effects, also thanks to the cosine decay.
>
> 4. "I understand that this paper focuses on learning the connectivity pattern from different inputs, like rgb and flow. Have the authors tried using RGB alone and searching the architecture?"
>
> Yes, we also tried the search while only using two RGB input stems. Our evolved RGB-only multi-stream model obtains 30.3% accuracy on the MiT dataset (with Table 3 setting) which is slightly lower than its RGB+Optical flow counterpart (31.4%). The hand-designed two-stream (late fusion) RGB-only model obtains 28.3%.

---

### Official Review · AnonReviewer3 · 2019-10-24
**Official Blind Review #3**

**Rating:** 8

**Review:**

This paper is a neural architecture search paper. In particular, it applies this to finding better neural architectures for video understanding, emphasizing exploring the video temporal resolutions needed and how to combine intermediate representations capturing appearance and motion. It introduces a somewhat new algorithm for connection-strength-weighted architecture evolution focused on this high-level information fusion problem of video understanding.

I am no expert in the problem domain of this paper (video understanding) but I fond the paper very clear and well-written and easy to understand. The techniques and thinking used seemed good (e.g., using dilated convolutions rather than manual preparing videos with different temporal resolutions!).

The evolutionary search algorithm was not wildly original or a huge breakthough in the general context of previous work, but seems appropriate, well thought out and works well.

The results reported are very strong. They get state-of-the-art results on two datasets. I particularly appreciated the evident care in producing strong baselines and ablations (their Charades 2-stream baseline also outperforms all previous work; they show the general strength of four-stream architectures and a random architecture with connection strength learning).

You mention at the start of section 4.2 that your models have the equivalent number of parameters to ResNet-50. This is good, but you should probably emphasize it more/earlier, since I'd been worrying that you were only winning due to size....

**Experience Assessment:**

I do not know much about this area.

**Review Assessment: Checking Correctness Of Derivations And Theory:**

I assessed the sensibility of the derivations and theory.

**Review Assessment: Checking Correctness Of Experiments:**

I assessed the sensibility of the experiments.

**Review Assessment: Thoroughness In Paper Reading:**

I made a quick assessment of this paper.

---

> ### Author Response · Authors · 2019-11-12
> **We thank the reviewer for the comments and for this suggestion.**
>
> "You mention at the start of section 4.2 that your models have the equivalent number of parameters to ResNet-50. This is good, but you should probably emphasize it more/earlier, since I'd been worrying that you were only winning due to size...."
>
> We thank the reviewer for the comments and for this suggestion. We will emphasize more that AssembleNet models use the equivalent number of parameters to previous models, introducing them early in the paper.

---

### Official Review · AnonReviewer4 · 2019-10-30
**Official Blind Review #4**

**Rating:** 6

**Review:**


Summary:

This paper aims to adapt the standard neural architecture search scheme to search a two-input convolutional neural network for video representations. To this end, the paper formulates a direct acyclic graph with two input nodes (for RGB image and optical flow), where each node represents some pre-composed layers and edge represents the data flow with a trainable weight. The searching policy is a modified evolutionary algorithm, which is guided by the trainable weights on the edge, and a set of graph limitations are in-place to avoid over-complicated graphs. The best-selected model outperforms previous baselines and achieves a new state-of-the-art on two video datasets.

Overall, this paper presents a concrete application of neural architecture search for video CNN with interesting results. Edge-weight guided evolutionary algorithm also demonstrates a small improvement in the ablation study. My concern, as detailed later, is if the comparison only with previously human-designed models is necessary. Nevertheless, this paper presents an interesting application of NAS and discover a feasible way to conduct an evolutionary algorithm within a reasonable cost (less than 100 sampled architectures).


Strength:
+ Writing in good shape, easy to follow and understand.
+ Motivation is clear and timely, reformulate neural architecture search for video representation is novel.
+ Clear experimental settings and reasonable convincing results.


Weakness:
- Lack of comparison with previous neural architecture search algorithms
Although the results yield that the proposed new search space is meaningful, considering each model has a similar dimension comparing to ResNet-50, it is still unknown if only comparing to human-designed model is a truly fair baseline. In my perspective, since this paper is built on top of NAS strategy with minor adaptation, could the author add one comparison experiment that, the proposed new search space is superior to those previous NAS spaces? For example, one could based on the earlier two-stream ResNet-50 with RGB+F modality, switch the backbone model into a NAS-based one and search with the same evolutionary algorithm (removing the edge weights adaptation). Otherwise, the improvement shown in the paper is not that surprising.


**Experience Assessment:**

I have published one or two papers in this area.

**Review Assessment: Checking Correctness Of Derivations And Theory:**

N/A

**Review Assessment: Checking Correctness Of Experiments:**

I carefully checked the experiments.

**Review Assessment: Thoroughness In Paper Reading:**

I read the paper thoroughly.

---

> ### Author Response · Authors · 2019-11-12
> **We thank the reviewer for the comments, questions and suggestions.**
>
> We thank the reviewer for the comments, questions and suggestions. The main contribution of this paper is in the introduction of the multi-stream (e.g., four-stream) architecture with evolved connectivity for videos. Previous NAS works focused on discovering modules to be used within a single-stream architecture (due to being designed mostly for images), while our search focuses on newly exploring multi-stream networks for videos and their stream connectivity, which we believe are two complementary directions. In the future we could combine our connectivity search with the previous-NAS-like module-level search suggested by the reviewer, jointly obtaining more sophisticated networks with evolved connectivity between complex modules, which will likely bring performance gains but will require a larger search space. Also note that all our models and hand-designed baselines used in the experiments are composed of the exact same convolutional modules and only their connectivity differ.

---

> > ### Comment · AnonReviewer4 · 2019-11-15
> > **acknowledged**
> >
> > Thanks for the reply. Nevertheless, I think the NAS-baseline I proposed will further evidence the contribution but does not take away your contribution.

---

### Decision · Program_Chairs · 2019-12-19

**Decision:**

Accept (Poster)

**Comment:**

The submission applies architecture search to find effective architectures for video classification. The work is not terribly innovative, but the results are good. All reviewers recommend accepting the paper.

---

> ### Author Response · Authors · 2019-12-22
> **We thank the reviewers and the AC for reviewing**
>
> We are very pleased that our paper is accepted to ICLR 2020.
>
> However, we find the AC's short comment "The work is not terribly innovative" provided without any explanation confusing and misleading, as none of the reviewers expressed any concern regarding the novelty of our paper. One reviewer explicitly stated the novelty as the strength of the paper, and we are not aware of any previous neural architecture search works or video understanding works exploring multi-stream connectivity learning.